# Machine learning-based drought prediction using Palmer Drought Severity Index and TerraClimate data in Ethiopia

Tadele Melese[1,2]*, Gizachew Assefa,[3] Baye Terefe[4], Tatek Belay[5], Getachew Bayable[1,2], Abebe Senamew[1]

1 Department of Natural Resource Management, College of Agriculture and Environmental Science, Bahir Dar University, Bahir Dar, Ethiopia, 2 Department of Information System science, Faculty of Science and Engineering, Soka University, Tokyo, Japan, 3 Department of Land Administration and Survey, Institute of Land Administration and Survey, Debre Markos University, Debre Markos, Ethiopia, 4 Department of Geography, Injibara University, Injibara, Ethiopia, 5 Department of Geography, College of Social Science and Humanities, Debre Tabor University, Debre Tabor, Ethiopia

* tadelemelese21m@gmail.com

## Abstract

Accurate drought prediction is essential for proactive water management and agricultural planning, especially in regions like Ethiopia that are highly susceptible to climate variability. This study investigates the classification of the Palmer Drought Severity Index (PDSI) using machine learning models trained on TerraClimate data, incorporating variables such as precipitation, temperature, soil moisture, and vapor pressure deficit. We employed several classifiers Logistic Regression, Support Vector Machine (SVM), k-Nearest Neighbors (KNN), Decision Tree, Random Forest, Gradient Boosting, Naive Bayes, AdaBoost, and XGBoost with Logistic Regression serving as a baseline statistical approach for comparison. To address data imbalance across drought classes, we applied a hybrid resampling method combining manual upsampling and SMOTE. Hyperparameter tuning was conducted using grid search and cross-validation. Random Forest outperformed all models, achieving an accuracy of 71.18%, F1-score of 0.71, and ROC AUC of 0.9000. Gradient Boosting and SVM also performed well with ROC AUC values of 0.8982 and 0.8681, respectively. SHAP analysis revealed that soil moisture, precipitation, and vapor pressure deficit were the most influential features in predicting drought severity. For benchmarking, an ARIMA (3,1,2) time-series model was applied but yielded poor performance (RMSE = 1.789, R² = −0.077), confirming the advantages of non-linear machine learning techniques for complex climate data. The results highlight the utility of ensemble learning in environmental modelling, offering valuable insights for drought early warning systems and climate resilience planning in Ethiopia. Future work should explore integrating localized predictors and real-time data to enhance prediction robustness.

**Data availability statement:** The data underlying the findings of this study are publicly available. The climate and hydrological variables were obtained from the TerraClimate dataset via Google Earth Engine (https://earthengine.google.com). The full code and preprocessed data used to replicate the study results are available on GitHub: https://github.com/tadele-melese/PDSI_Prediction.

**Funding:** The author(s) received no specific funding for this work.

**Competing interests:** The authors have declared that no competing interests exist.

# 1. Introduction

Drought is a severe and widespread natural disaster that significantly impacts agriculture, water resources, and the economy. It affects millions of people around the world, particularly in regions where agriculture forms the backbone of livelihoods. Researchers have developed mechanism and tools to monitor drought. Among the tools used to monitor and quantify drought, the Palmer Drought Severity Index (PDSI) stands out as a robust metric that integrates both meteorological and hydrological data to assess drought conditions. PDSI has been widely adopted in various regions globally, but its application in Sub-Saharan Africa, particularly Ethiopia, remains under-explored [1–3]. Ethiopia, a country highly dependent on rain-fed agriculture, is particularly vulnerable to the adverse effects of drought. In recent years, climate variability has intensified the frequency and severity of droughts, posing significant challenges to food security and water availability in the region [4].

Traditional drought prediction methods in Ethiopia rely on historical data and statistical approaches, but often struggle to capture complex, non-linear relationships between environmental variables [5–7]. These methods often lack the ability to integrate multiple data sources, like remote sensing data, which can provide a more comprehensive understanding of drought conditions on large spatial scales [8–11]. The need for advanced models to predict drought conditions is urgent, enabling better preparedness and response strategies. Advancements in remote sensing technologies and high-resolution climate data have opened new possibilities for drought monitoring and prediction. TerraClimate, a global dataset with monthly climate and water balance variables, provides a valuable source for drought prediction [12,13]. Combining machine learning techniques with large datasets can significantly improve the accuracy and timeliness of drought predictions.

Machine learning is a powerful tool in environmental modelling, capable of processing large data volumes and identifying hidden patterns. It can be used for drought prediction, recognizing complex interactions between climatic variables like precipitation, temperature, and soil moisture, which are crucial in determining drought severity [14,15]. The study aims to utilize machine learning techniques to predict the PDSI using TerraClimate data, specifically targeting Ethiopia. Ethiopia's diverse topography and climate variability make it an ideal case study for machine learning in drought prediction. Despite abundant data, there is limited research on advanced machine learning techniques for predicting drought in Ethiopia. This study aims to fill this gap by developing and evaluating models that accurately predict PDSI, providing a valuable tool for policymakers and practitioners in the region [16,17].

This study uses high-resolution TerraClimate data and advanced machine learning algorithms to predict drought conditions in Ethiopia. It uses modern data science techniques to improve drought prediction accuracy. The study incorporates a wide range of climatic variables from the TerraClimate dataset, accounting for meteorological and hydrological factors. Machine learning models identify complex, non-linear relationships between variables, which are often overlooked in traditional models [18–20]. The study offers valuable technical and practical insights into accurate drought predictions, enabling early warning systems and timely interventions to

mitigate the impact of drought on vulnerable populations [21]. The study suggests that predicting drought conditions more accurately in Ethiopia could significantly improve food security and economic stability, potentially contributing to global efforts to enhance drought resilience, and potentially applicable to other regions with similar climates.

In a nutshell, this study represents a significant advancement in the field of drought prediction, combining state-of-the-art machine learning techniques with high-resolution climate data to develop a robust model for predicting the PDSI in Ethiopia. By addressing the limitations of traditional methods and using the power of modern data science, this research has the potential to significantly improve our ability to predict and respond to droughts, both in Ethiopia and beyond.

## 2. Study area description

The study area encompasses Ethiopia (Fig 1), situated between latitudes 3° and 15°N and longitudes 33° and 48°E in the Horn of Africa. Ethiopia's diverse topography, ranging from rugged highlands above 4,500 meters to lowland plains as low

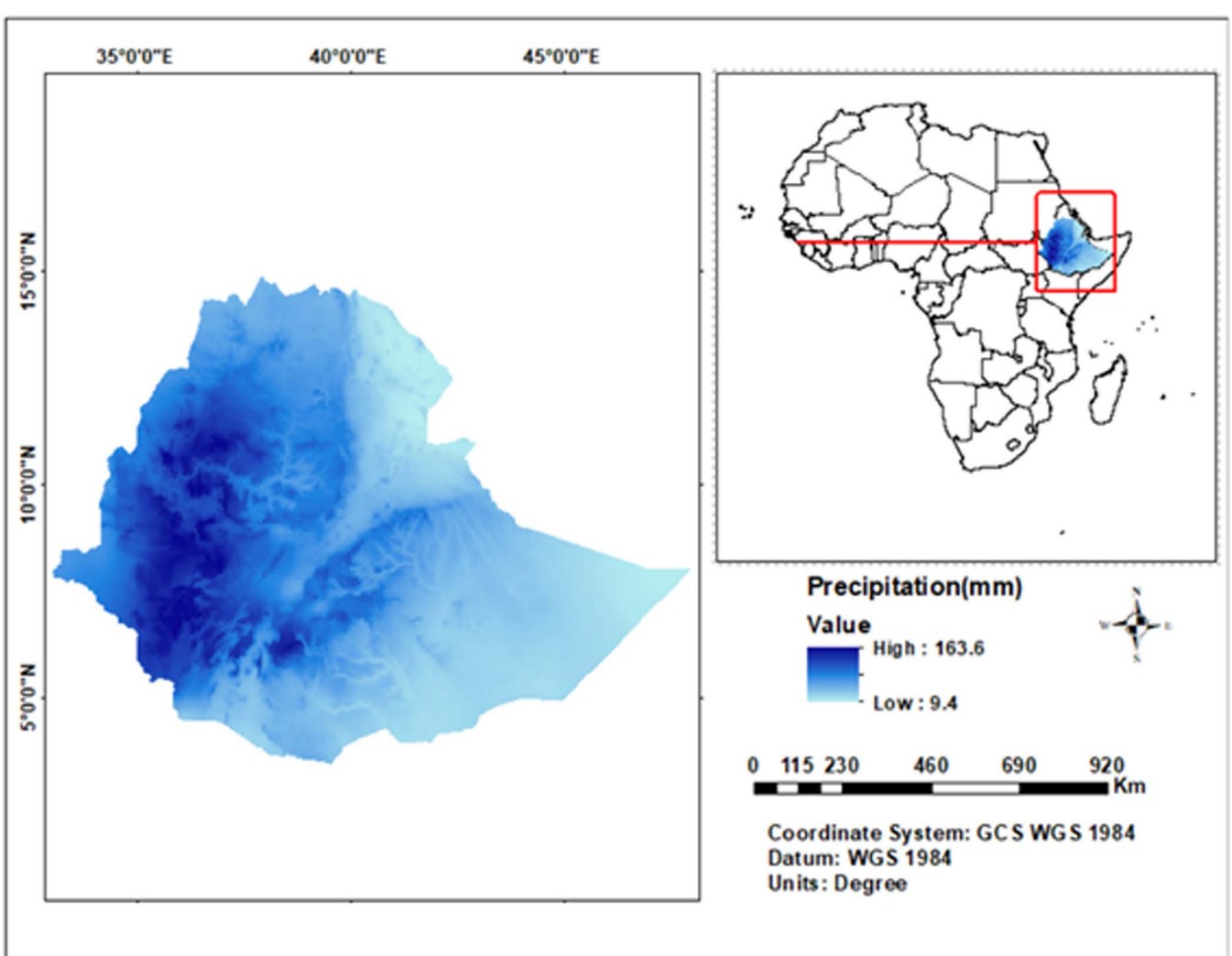

**Fig 1. Location map of Ethiopia.** The map was generated by the authors using freely available shapefile and rainfall data from the TerraClimate dataset (https://www.climatologylab.org/terraclimate.html), which is openly accessible and can be used under open-access conditions.

as 100 meters, significantly influences its climatic conditions. Ethiopia's climate is influenced by its proximity to the Equator, the Intertropical Convergence Zone, and seasonal monsoon systems, resulting in varied precipitation patterns [22]. The central and northern highlands, which house most of the population, enjoy cool temperatures and significant rainfall during the main rainy season (June to September). In contrast, the eastern and southern lowlands experience hotter, drier conditions, where rainfall is scarce and irregular. Ethiopia's diverse climate presents challenges for drought prediction due to its historical vulnerability to extreme weather events.

## 3. Materials and methods

### 3.1. Data collection

**3.1.1. PDSI data.** This study uses PDSI data from TerraClimate satellite data, a standardized measure of drought, to assess its severity and duration across Ethiopia. The data, covering the period from 1981 to 2023, provides a detailed view of drought conditions across diverse climatic regions. Before analysis, the data undergoes preprocessing steps to ensure quality and consistency, including handling missing values, smoothing temporal sequence inconsistencies, and aligning the data with TerraClimate data's spatial resolution. The dataset reveals a range of drought and wet conditions, with Near Normal as the most frequent category (21.32%). Mild Drought, Moderate Drought, and Incipient Dry Spell are prevalent, followed by Slightly Wet and Incipient Wet Spell. Extreme conditions like Severe Drought and Moderate Wet are less frequent, with 5.62% and 3.29% respectively. Extreme Drought is observed in 0.97% of the dataset, while Very Wet is just 0.39%. The limited presence of Extremely Wet conditions in the data highlights the importance of understanding drought dynamics and their impacts (Fig 2).

**3.1.2. TerraClimate data.** The TerraClimate dataset offers global, monthly gridded data at a high spatial resolution of 1/24° (~4-km), covering climate and water balance variables from 1958 to the present [12] that are essential for analysing drought dynamics and their impact on the PDSI. The dataset includes a range of climate and water balance variables essential for analysing drought dynamics (see Table 1 for variable descriptions, units, and scales). The study utilized key variables include precipitation (pr), maximum temperature (tmmx), and minimum temperature (tmmn), which provide insights into the region's hydrological and thermal conditions. Potential evapotranspiration (pet) and actual evapotranspiration (aet) are included to evaluate water demand and actual water loss from the surface, while soil moisture (soil) helps to assess water availability in the soil. Solar radiation (srad) and vapor pressure (vap) offer information on energy fluxes and atmospheric moisture content, respectively. Water runoff (ro) reflects excess water flow, while wind

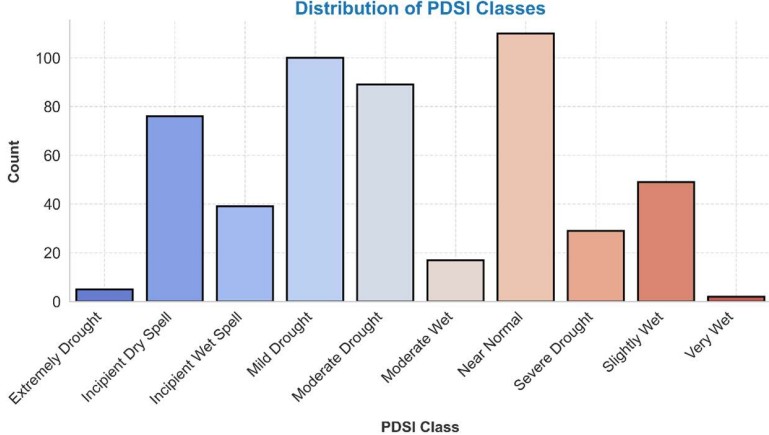

**Fig 2. Percentage distribution of PDSI classes.**

**Table 1. TerraClimate dataset variables, including names, descriptions, minimum and maximum values, units, and scales.**

| Name | Description | Min | Max | Units | Scale |
|------|-------------|-----|-----|-------|-------|
| aet | Actual evapotranspiration derived using a one-dimensional soil water balance model | 0 | 3140 | mm | 0.1 |
| def | Climate water deficit, derived using a one-dimensional soil water balance model | 0 | 4548 | mm | 0.1 |
| pdsi | Palmer Drought Severity Index | −4317 | 3418 | | 0.01 |
| pet | Reference evapotranspiration (ASCE Penman-Montieth) | 0 | 4548 | mm | 0.1 |
| pr | Precipitation accumulation | 0 | 7245 | mm | 0 |
| ro | Runoff, derived using a one-dimensional soil water balance model | 0 | 12560 | mm | 0 |
| soil | Soil moisture, derived using a one-dimensional soil water balance model | 0 | 8882 | mm | 0.1 |
| srad | Downward surface shortwave radiation | 0 | 5477 | W/m² | 0.1 |
| swe | Snow water equivalent, derived using a one-dimensional soil water balance model | 0 | 32767 | mm | 0 |
| tmmn | Minimum temperature | −770 | 387 | °C | 0.1 |
| tmmx | Maximum temperature | −670 | 576 | °C | 0.1 |
| vap | Vapor pressure | 0 | 14749 | kPa | 0.001 |
| vpd | Vapor pressure deficit | 0 | 1113 | kPa | 0.01 |
| vs | Wind-speed at 10m | 0 | 2923 | m/s | 0.01 |

speed (vs) and vapor pressure deficit (vpd) are indicators of atmospheric conditions influencing evapotranspiration. Snow water equivalent derived using a one-dimensional soil water balance model (swe) is not considered due to missing data problem. The analysis of these variables aids in evaluating the spatial and temporal dynamics of drought and its impact on PDSI, providing a comprehensive understanding of the region's hydrological and thermal conditions.

In this study, we extracted the climate data using the TerraClimate dataset available on Google Earth Engine (GEE) (ID: IDAHO_EPSCOR/TERRACLIMATE) in https://earthengine.google.com/. This dataset provides high-resolution (approximately 4-km) monthly climate and climatic water balance data. We followed these key steps for data extraction through GEE. We used the IDAHO_EPSCOR/TERRACLIMATE dataset, which provides monthly data on variables such as pr, tmmx, tmmn, and soil and others that were employed in this study. We defined the geographic extent of our study area using a shapefile (uploaded to GEE Assets) and clipped the data accordingly. We filtered the TerraClimate image collection by date to match the study period. For each selected variable, we extracted relevant bands and computed spatial means over the study area to generate time series data. The processed data were exported as CSV files from GEE to Google Drive and subsequently used for model training and evaluation (Fig 3).

### 3.2. Data preprocessing

**3.2.1. Exploratory Data Analysis (EDA).** A comprehensive EDA was conducted to explore the distribution, trends, and relationships within the dataset, focusing on both the input features and the target variable, PDSI. Histograms were used to examine the distribution of climate variables and the PDSI, revealing that some variables, including aet, precipitation, soil moisture, minimum temperature, and vapor pressure exhibited non-normal distributions with skewness and potential outliers (see Appendix). Time series trend analysis showed distinct seasonal patterns in variables such as temperature and precipitation, with the PDSI also displaying periodic fluctuations that corresponded with these seasonal changes (see Fig 4). Understanding these temporal patterns was crucial for capturing the dynamic nature of droughts in the model. Time series analysis also helped in identifying missing data points across the dataset. These gaps, including those in the PDSI series, were handled using linear interpolation to maintain the integrity of the time series data. This approach ensured that the temporal continuity of the dataset was preserved, which is crucial for accurate time series modelling. The code used in this study is available at the following GitHub repository: https://github.com/tadele-melese/PDSI_Prediction.

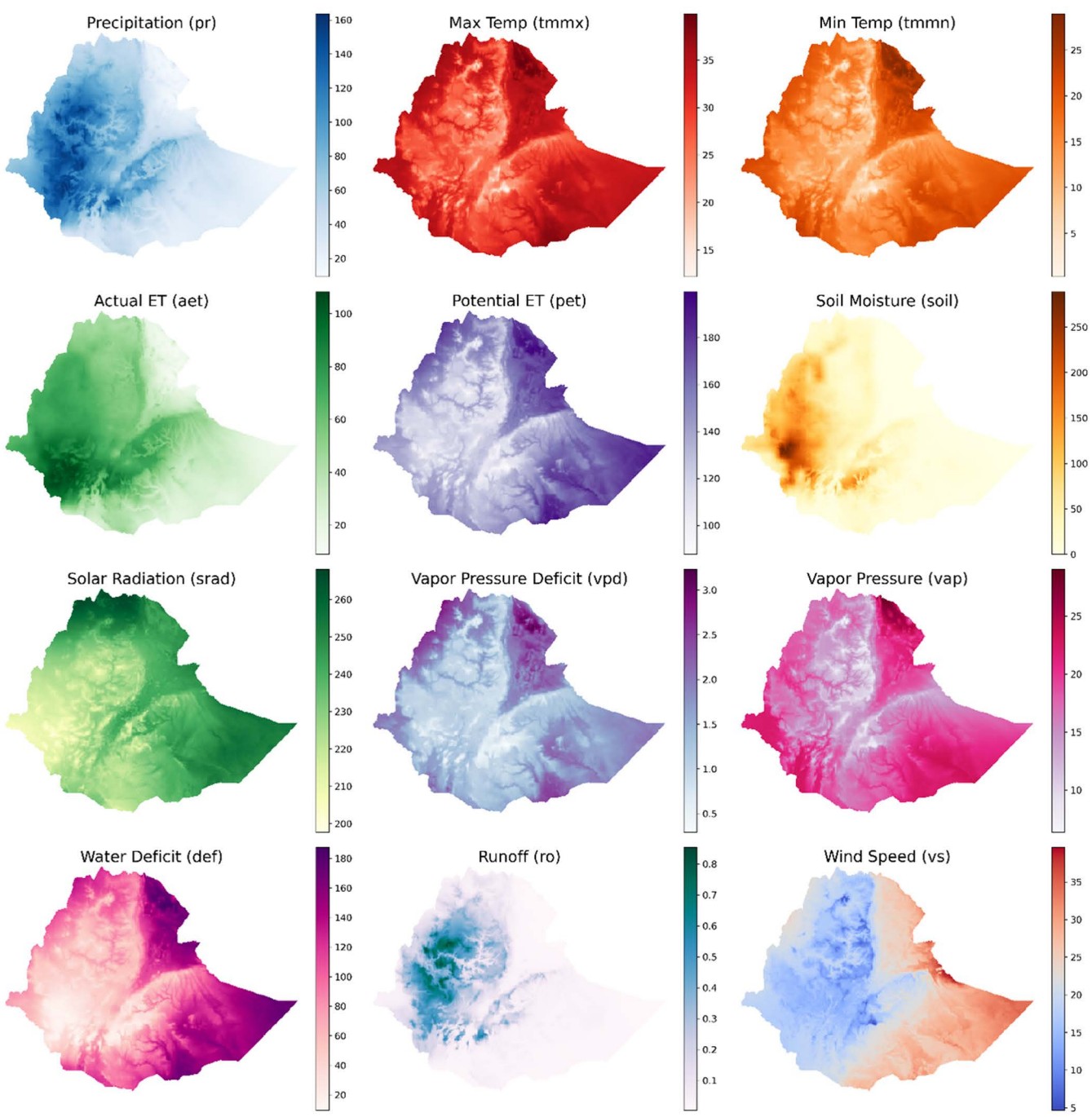

**Fig 3. This figure displays twelve climate variables for Ethiopia, each represented with a distinct color-coded map.** The variables include Precipitation (pr), showing total monthly precipitation in shades of blue; Maximum Temperature (tmmx), indicating the highest temperatures recorded each month with varying reds; Minimum Temperature (tmmn), depicting the lowest temperatures in different shades of orange; Actual Evapotranspiration (aet), reflecting water lost through evaporation and transpiration in greens; Potential Evapotranspiration (pet), representing potential water loss in purples; Soil Moisture (soil), mapping moisture levels in soil with browns; Solar Radiation (srad), showing intensity of solar radiation in yellows and greens; Vapor Pressure Deficit (vpd), illustrating air dryness in blues and purples; Vapor Pressure (vap), displaying vapor pressure in shades of blue; Drought Evapotranspiration (def), representing drought-related evapotranspiration in reds; Runoff (ro), showing runoff levels in shades of blue; and Vegetation Index (vs), depicting vegetation coverage in greens. Each map features a color bar legend, grid lines, a scale bar for distance, and a north arrow for orientation.

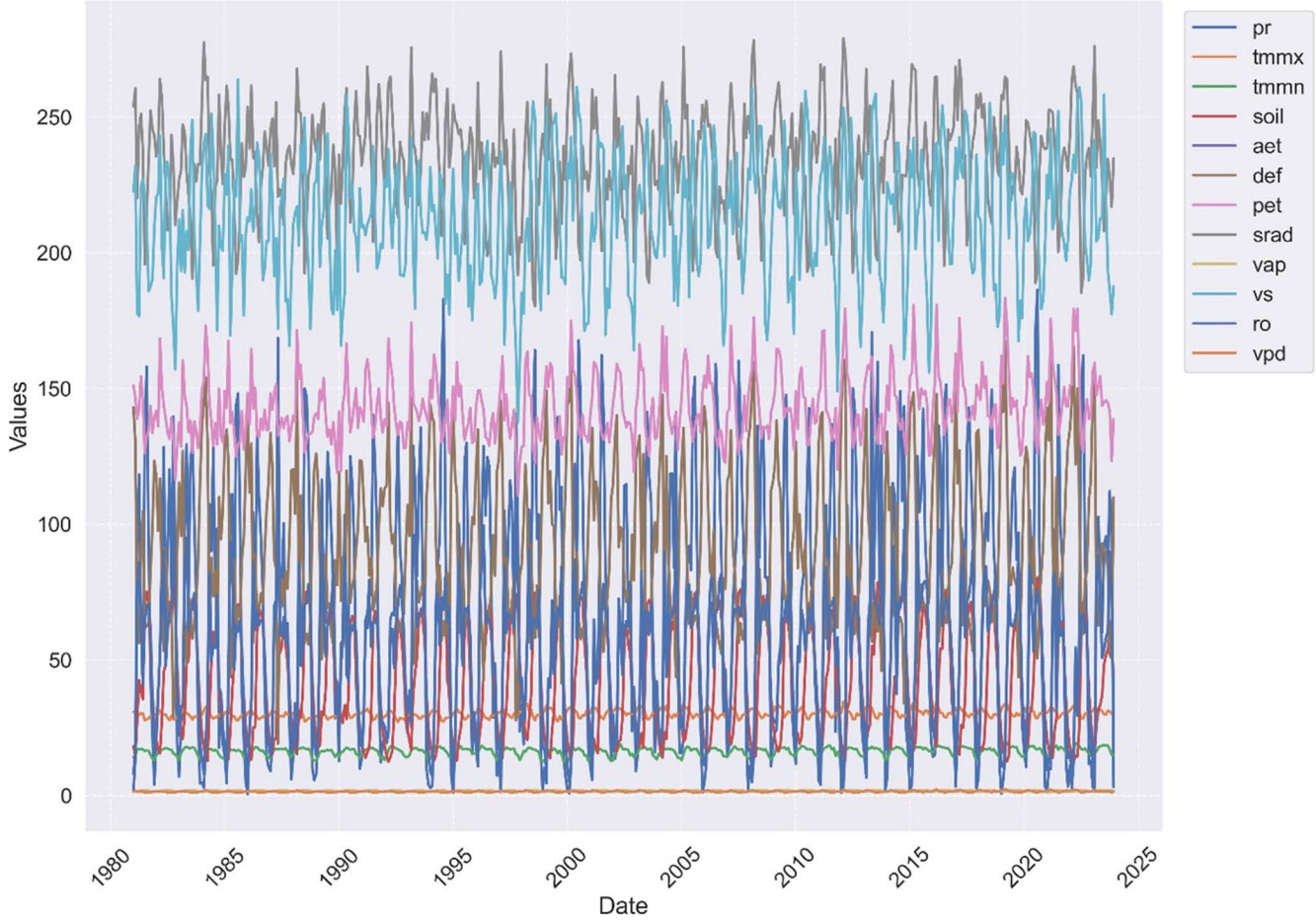

**Fig 4. Time series analysis of TerraClimate variables (e.g., precipitation, maximum temperature, soil moisture) from 1981 to 2023, illustrating seasonal trends and patterns over time.**

Cumulative Distribution Function (CDF) and Probability Density Function (PDF) plots were applied not only to the input variables (Appendix) but also to the target variable, PDSI. The CDF for PDSI highlighted the proportion of data points below certain threshold values, which is essential for understanding the likelihood of various drought severities (see Fig 5 right). The PDF for PDSI illustrated the probability of different PDSI values, revealing a tendency toward certain severity levels and the presence of extreme drought conditions in the dataset (see Fig 5 left).

Boxplots identified outliers in both the input variables and the PDSI. It used to identify and manage outliers, with comparisons before and after outlier removal shown in Fig 6. For instance, outliers were detected in PET and SRAD (see Fig 4). These outliers were managed using a capping technique, where extreme values were adjusted to reduce their influence on the model while preserving the overall distribution.

Correlation analysis, presented through a heatmap, revealed significant relationships among the input variables and between some inputs and the PDSI (see Fig 7). For example, strong correlations were observed between maximum temperature (Tmax) and PET, while weaker correlations with PDSI indicated that multiple variables contribute uniquely to drought prediction. By thoroughly exploring the dataset through EDA, including detailed analysis of the PDSI as the target

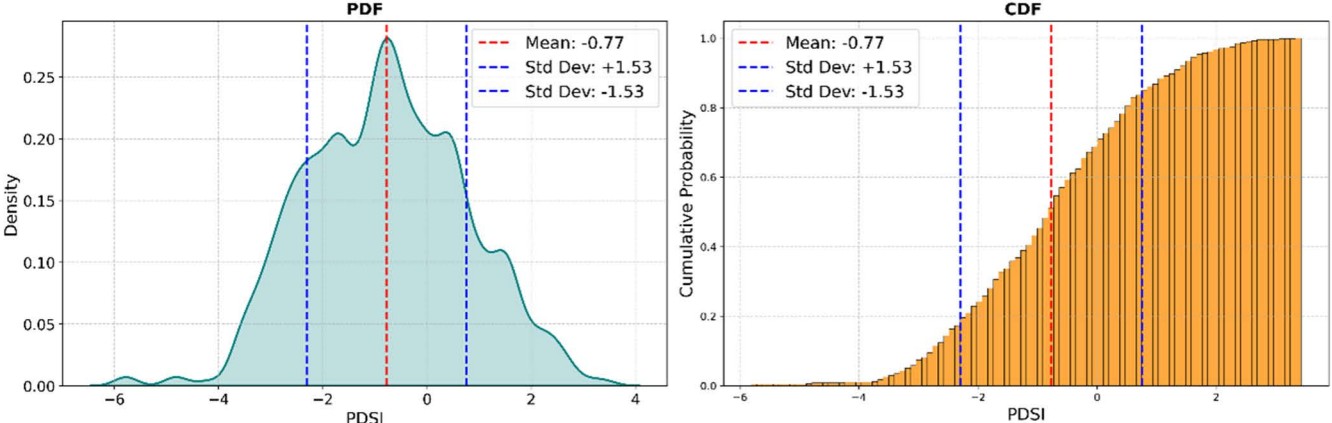

**Fig 5. Distribution analysis of PDSI from 1981 to 2023, featuring the Probability Density Function (PDF, left) and Cumulative Distribution Function (CDF, right).** The red dashed line indicates the mean PDSI (−0.12), with blue dashed lines showing one standard deviation (±1.85).

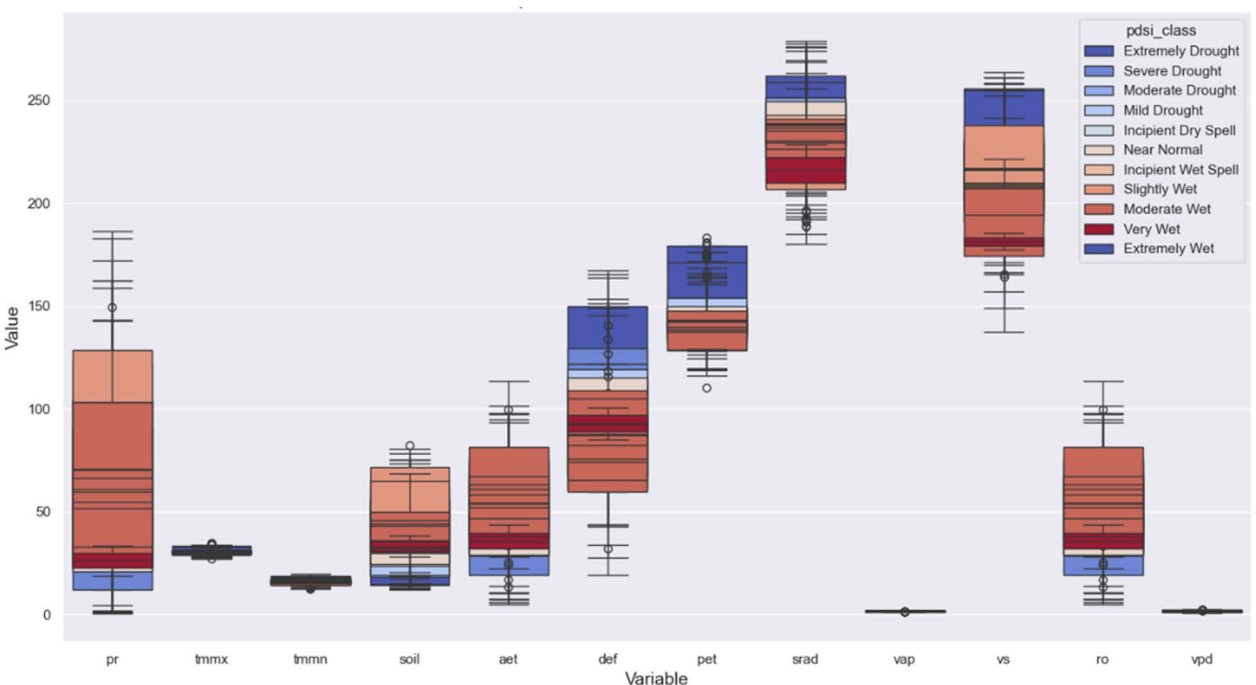

**Fig 6. Comparison of boxplots for TerraClimate variables (e.g., PET, SRAD) before (left) and after (right) outlier removal, showing reduced variability post-capping of extreme values.**

variable, we ensured that the data preprocessing steps were well-informed, setting a strong foundation for the subsequent machine learning tasks.

In addressing class imbalance within the dataset, a two-step approach was implemented. Initially, to address the under-representation of certain classes, manual duplication was employed. Specifically, the minority classes were identified, and

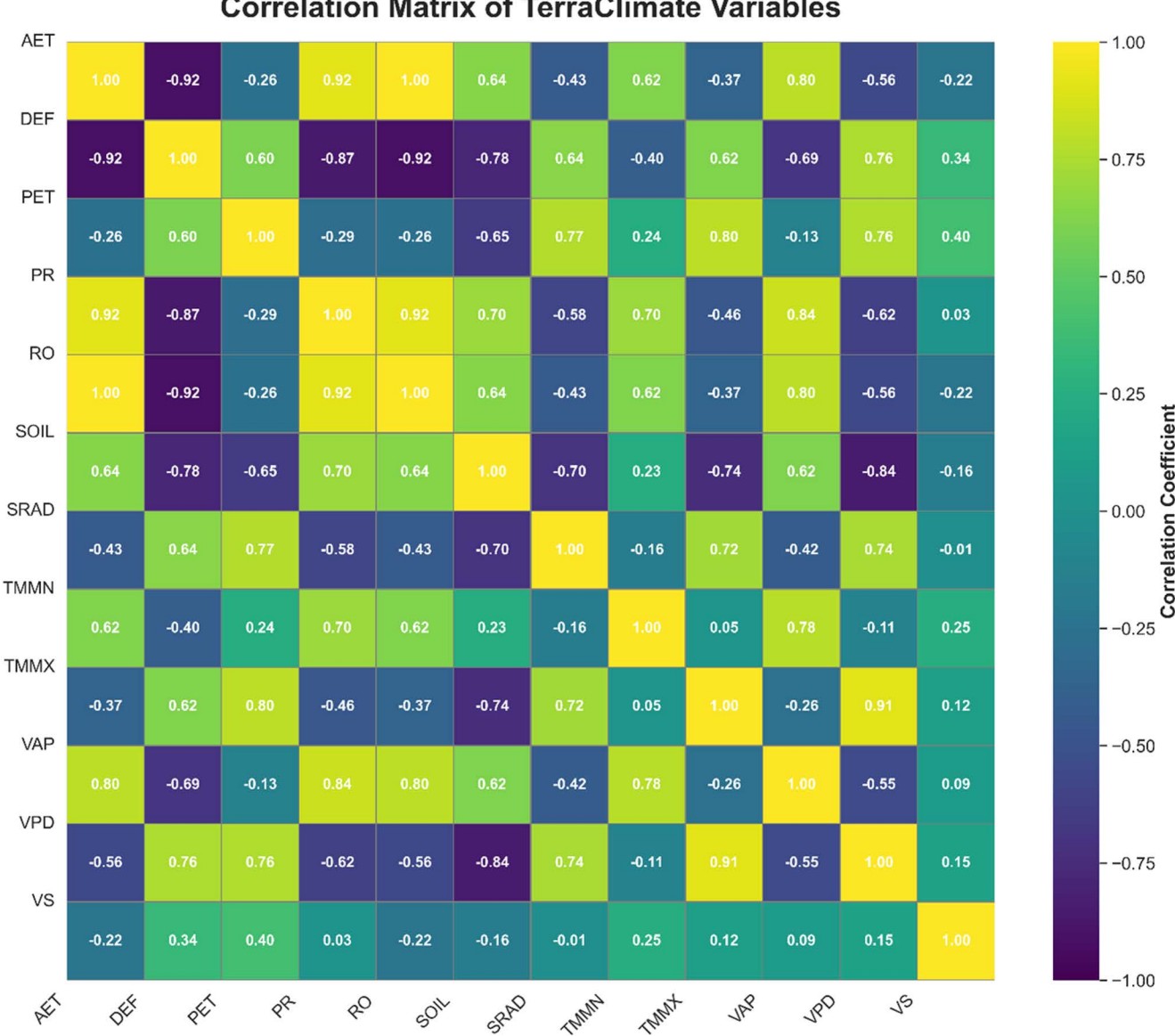

**Fig 7. This heatmap illustrates the pairwise correlations between different TerraClimate variables, with correlation coefficients ranging from −1 (strong negative correlation) to 1 (strong positive correlation.**

samples from these classes were duplicated to achieve a target count of 10 samples per class. This manual adjustment aimed to increase the representation of the minority classes within the dataset.

Subsequently, to further balance the dataset, Synthetic Minority Over-sampling Technique (SMOTE) was applied. SMOTE is a technique used to generate synthetic samples for the minority classes to achieve a balanced dataset. In this process, the features and target variables were first separated from the dataset, excluding non-essential columns. SMOTE was then applied to the remaining data with a specified number of neighbours generating synthetic examples to enhance the representation of the minority classes. This combination of manual duplication and SMOTE ensured that

the dataset was adequately balanced, facilitating more effective training and evaluation of machine learning models. To address class imbalance, we combined manual duplication of minority classes (e.g., Very Wet, Extreme Drought) with SMOTE to avoid over-reliance on synthetic data, which could introduce artificial patterns. We tested class-weighting as an alternative but found SMOTE more effective for this dataset. Future work could explore cost-sensitive learning to further refine performance.

To optimize the performance of machine learning classification models, feature standardization was applied to the input variables. This involved cantering each feature by subtracting its mean and scaling it by its standard deviation, resulting in a distribution with a mean of zero and a standard deviation of one. Standardization is particularly important for algorithms that are sensitive to feature scaling, such as Support Vector Machines (SVM) and logistic regression [23–25]. Lag features were created to improve predictive accuracy by capturing temporal dependencies. These features account for the effects of historical climate conditions on current PDSI levels. Lag features were generated for lead times of 1 month, 3 months, 6 months, and 12 months to assess the impact of varying temporal scales on drought prediction (Fig 8).

### 3.3. Machine learning models for classification

**3.3.1. Model selection.** In this study, we selected a variety of machine learning classifiers to address the classification of PDSI data. Model selection was guided by the diversity of algorithmic approaches and their proven efficacy in environmental modelling. The chosen models included Logistic Regression, Random Forest, Support Vector Machine (SVM) [26], K-Nearest Neighbors (KNN), Gradient Boosting, Naive Bayes, and Decision Tree [27]. Logistic Regression was chosen for its simplicity and interpretable base line, providing a probabilistic approach to class membership. Random Forest was selected for its ensemble method that combines multiple decision trees to enhance classification accuracy and robustness [28]. SVM was included due to its effectiveness in handling complex decision boundaries using various kernel

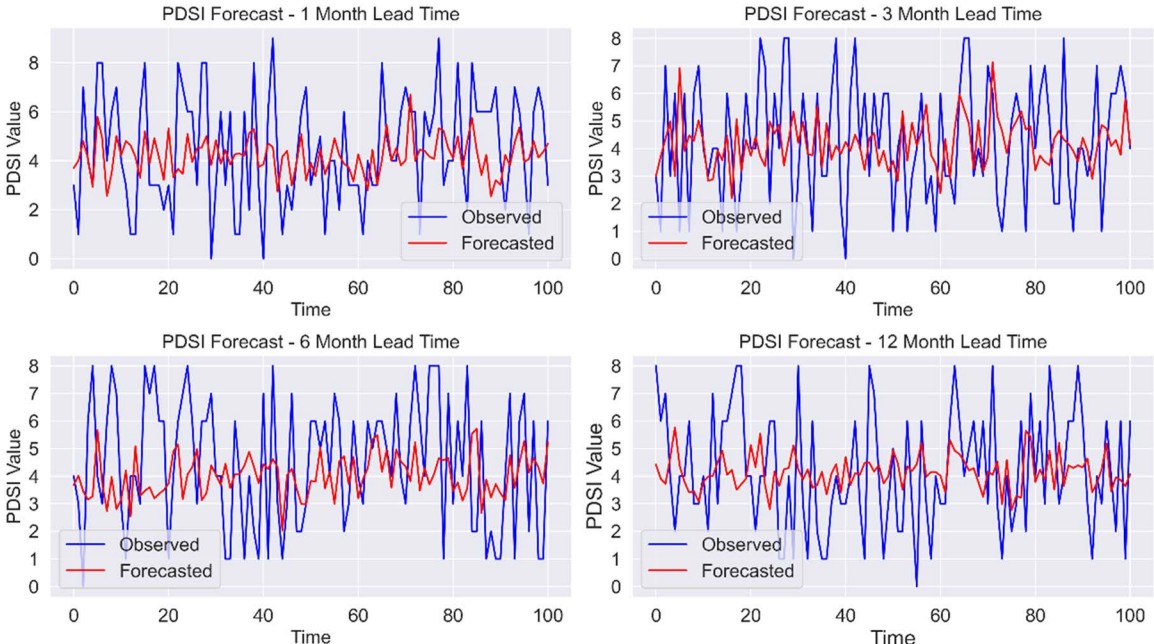

**Fig 8. Observed vs. Forecasted PDSI for Different Lead Times.** This figure shows the performance of PDSI forecasting with various lead times: 1 month, 3 months, 6 months, and 12 months. For each subplot, the blue line represents the observed PDSI values, while the red line represents the forecasted values. The x-axis shows the date in Year-Month format, and the y-axis shows the PDSI values. The forecast accuracy may vary depending on the lead time, providing insights into the effectiveness of forecasting models over different time horizons.

functions. KNN was used for its non-parametric approach, which classifies data based on the majority vote of nearest neighbors, capturing local patterns. Gradient Boosting and AdaBoost were selected for their ensemble techniques that improve classification performance by aggregating multiple weak classifiers AdaBoost [29,30]. Naive Bayes was chosen for its efficiency and simplicity, based on Bayes' theorem with strong independence assumptions. Finally, Decision Tree was included for its ability to provide clear, interpretable decision rules by splitting data based on feature values. Deep learning models (e.g., CNNs, LSTMs) were not included due to limited computational resources and the adequacy of tabular data for this study, though their potential is noted for future work.

**3.3.2. Model training.** The machine learning classifiers were trained on PDSI data using a well-defined training set. The process involved fitting the models to the data, learning to map input features to target categories. Cross-validation techniques were used to evaluate the model's performance on different subsets of the data, providing a robust measure of its generalization ability. Hyperparameter tuning was performed after initial training to refine models and improve classification results. This involved adjusting parameters to enhance performance and prevent overfitting. Key parameters like regularization strength were optimized for Logistic Regression, while Random Forest models had their number of trees and maximum depth tuned to improve accuracy and robustness. For SVM, parameters including the kernel type and regularization parameter were adjusted to effectively manage complex decision boundaries. KNN's performance was enhanced by tuning the number of neighbors and distance metrics to better capture local patterns. Gradient Boosting and AdaBoost models were refined by adjusting the learning rates and number of boosting stages to maximize classification accuracy while avoiding overfitting. Naive Bayes had its smoothing parameters tuned to improve robustness against noisy data, and Decision Trees were optimized by adjusting parameters such as maximum depth and minimum samples per leaf to ensure clear and effective decision rules. This tuning process aimed to achieve optimal model performance and reliable classification of PDSI categories.

**3.3.3. Performance evaluation metrics.** To evaluate the performance of the classifiers, a range of metrics was employed to provide a comprehensive assessment. Accuracy was determined by the ratio of correctly classified instances to the total number of instances:

$$\text{Accuracy} = \frac{\text{TP} + \text{TN}}{\text{TP} + \text{TN} + \text{FP} + \text{FN}}$$

where (TP) represents true positives, (TN) true negatives, (FP) false positives, and (FN) false negatives. Precision was calculated as the proportion of true positives out of the total predicted positives:

$$\text{Precision} = \frac{\text{TP}}{\text{TP} + \text{FP}}$$

Recall, also known as Sensitivity, measured the proportion of true positives out of the total actual positives:

$$\text{Recall} = \frac{\text{TP}}{\text{TP} + \text{FN}}$$

The F1 Score combined Precision and Recall into a single metric, providing a balance between them:

$$\text{F1 Score} = \frac{\text{Precision*Recall}}{\text{T Precision} + \text{Recall}}$$

The Confusion Matrix offered a detailed breakdown of classification performance, displaying counts of true positives, true negatives, false positives, and false negatives, which helped in understanding the types of errors made

by each model. The ROC Curve (Receiver Operating Characteristic Curve) illustrated the trade-offs between the true positive rate (Recall) and the false positive rate across different threshold values. The AUC Score (Area Under the ROC Curve) quantified the model's ability to distinguish between classes, with a value of 1 indicating perfect discrimination and 0.5 indicating random guessing. These evaluations were conducted using Python, leveraging libraries such as scikit-learn, pandas, and NumPy for metric calculations and ROC Curve plotting. Google Earth Engine (GEE) was utilized to process and export TerraClimate satellite data, streamlining the data preparation phase and enabling a focus on model training and performance evaluation. This integrated approach ensured a thorough comparison of the classifiers, guiding the selection of the most effective model for predicting drought conditions in Ethiopia.

## 4. Results and discussion

### 4.1. Results

#### 4.1.1. Model performance evaluation.
Our experiment aimed to predict the PDSI classes using a set of machine learning classifiers, with preprocessing steps playing a crucial role in improving the model's performance. The data, which included various climate variables, underwent several transformations to ensure optimal performance of the classifiers. Initially, the dataset was cleaned and standardized using standard scaler, which adjusted the features to have a mean of 0 and a standard deviation of 1. This step is particularly important for machine learning models like SVM and KNN, which are sensitive to the scale of input features. Standardization ensures that each feature contributes equally to the learning process.

A critical challenge in the dataset was its imbalance, with certain PDSI classes being underrepresented. Imbalanced data often leads to biased models that perform poorly on minority classes. To address this, we employed a hybrid resampling technique that combined manual upsampling of the minority classes specifically (Very Wet, Extremely Drought and Near Normal), with SMOTE. SMOTE generates synthetic samples for the minority classes by interpolating between existing samples, which helps improve class representation without simply duplicating data points. This approach aimed to create a more balanced dataset, which would help the models learn to predict all classes more effectively, particularly those with fewer samples.

After resampling, the data was split into training and testing sets. Several machine learning classifiers were then trained, including Logistic Regression, Random Forest, SVM, KNN, Gradient Boosting, AdaBoost, Naive Bayes, and Decision Trees. We also incorporated XGBoost where applicable, given its reputation for strong performance in classification problems. Each classifier was evaluated based on its ability to predict PDSI classes, with performance metrics such as accuracy, precision, recall, F1-score, confusion matrix, ROC curves, and AUC scores.

Among the models, Random Forest emerged as the top performer, achieving an accuracy of 69.2%. This can be attributed to Random Forest's ability to handle imbalanced datasets better than many other classifiers due to its ensemble nature, which combines multiple decision trees to improve predictive accuracy and control overfitting. Additionally, Random Forest is robust to noise and performs well even when some classes have fewer instances. In contrast, models like SVM and AdaBoost struggled significantly, with SVM showing a very low accuracy of 23.2%, and AdaBoost achieving only 27.8%. These models found it challenging to predict the minority classes accurately, likely due to their sensitivity to data imbalance and feature scaling.

The precision, recall, and F1-scores for each model provided deeper insights into how well the classifiers performed on individual PDSI classes. For instance, Logistic Regression, despite being a simpler model, struggled to predict certain minority classes, resulting in undefined precision metrics for these labels. This revealed a key limitation in its ability to generalize across all PDSI categories. On the other hand, while Gradient Boosting and XGBoost delivered solid overall performance, they still encountered difficulties in distinguishing between the minority and majority classes, particularly when class distributions were highly skewed.

 

This experiment highlighted the complexity of predicting PDSI values using machine learning classifiers, especially when dealing with imbalanced datasets. The resampling techniques, while helpful, did not entirely solve the problem, indicating that further exploration of techniques like ensemble learning, hyperparameter tuning, might be necessary to improve classification across all classes. The poor performance of models like SVM and AdaBoost also suggests that certain classifiers may not be well-suited for this type of problem without significant adjustments.

**4.1.2. Hyperparameter tuning.** Hyperparameter tuning was conducted for several classifiers to enhance the performance of the PDSI classification task. For each model, key parameters were adjusted to improve the accuracy and ROC-AUC scores, which are vital for assessing the classifiers, especially given the imbalanced dataset. ROC curves for multiple classifiers across PDSI classes are presented in Fig 9, illustrating their discriminative abilities. The Logistic Regression model achieved its best performance with the parameters of maximum depth of none and minimum samples split of 2, resulting in an accuracy of 40.40% and a ROC AUC of 0.7823 (Table 2).

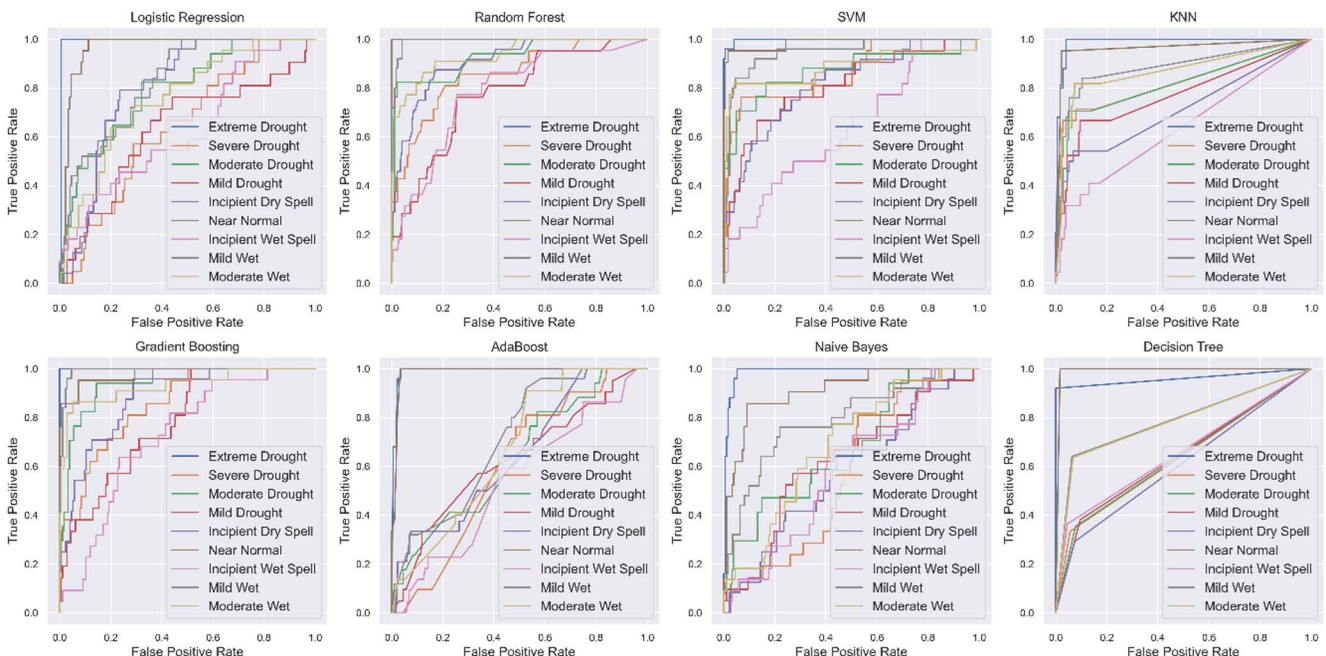

**Fig 9. ROC curves for multiple classifiers with different PDSI classes, illustrating how each model distinguishes between drought/wetness categories based on the ROC AUC performance.**

**Table 2. Classifier Performance Metrics After Hyperparameter Tuning.**

| Classifier | Precision | Recall | F1-Score | Accuracy | ROC AUC |
|---|---|---|---|---|---|
| Random Forest | 0.72 | 0.71 | 0.71 | 71.18 | 0.9000 |
| SVM | 0.67 | 0.66 | 0.66 | 66.16 | 0.8681 |
| KNN | 0.63 | 0.63 | 0.62 | 62.63 | 0.8295 |
| Gradient Boosting | 0.61 | 0.61 | 0.60 | 60.61 | 0.8982 |
| Logistic Regression | 0.41 | 0.40 | 0.40 | 40.40 | 0.7823 |
| Decision Tree | 0.56 | 0.56 | 0.55 | 55.56 | 0.7456 |
| AdaBoost | 0.34 | 0.33 | 0.32 | 33.33 | 0.7129 |
| Naive Bayes | 0.36 | 0.36 | 0.35 | 35.86 | 0.7140 |

Although this model did not provide the highest accuracy, its ROC AUC indicates a reasonable capability in distinguishing between classes. The Random Forest classifier, with the same hyperparameters, delivered a much better accuracy of 71.18% and an impressive ROC AUC of 0.90%, showing significant improvement over Logistic Regression. Similarly, the SVM model achieved an accuracy of 66.16% and a ROC AUC of 0.8681, indicating strong performance, though slightly lower than Random Forest.

The KNN algorithm reached an accuracy of 62.63% and a ROC AUC of 0.8295, showing a moderate balance between precision and sensitivity. Gradient Boosting, another ensemble method, yielded a slightly lower accuracy of 60.61%, but a ROC AUC of 0.8982, making it competitive in terms of classification capability. On the other hand, AdaBoost's performance was relatively poor, with an accuracy of just 33.33% and a ROC AUC of 0.7129. This model struggled with both accuracy and class separation. Naive Bayes also performed weakly, with an accuracy of 35.86% and a ROC AUC of 0.7140, though it did slightly better in terms of class distinction than AdaBoost. Finally, the Decision Tree model, with an accuracy of 55.56% and a ROC AUC of 0.7456, performed decently, though it was outperformed by more complex models like Random Forest and Gradient Boosting. This comparison underlines how hyperparameter tuning can greatly influence model performance, and ensemble models such as Random Forest and Gradient Boosting generally offered superior results for PDSI classification. A comparison of ROC AUC scores across all models is shown in Fig 10, highlighting the superior performance of Random Forest and Gradient Boosting.

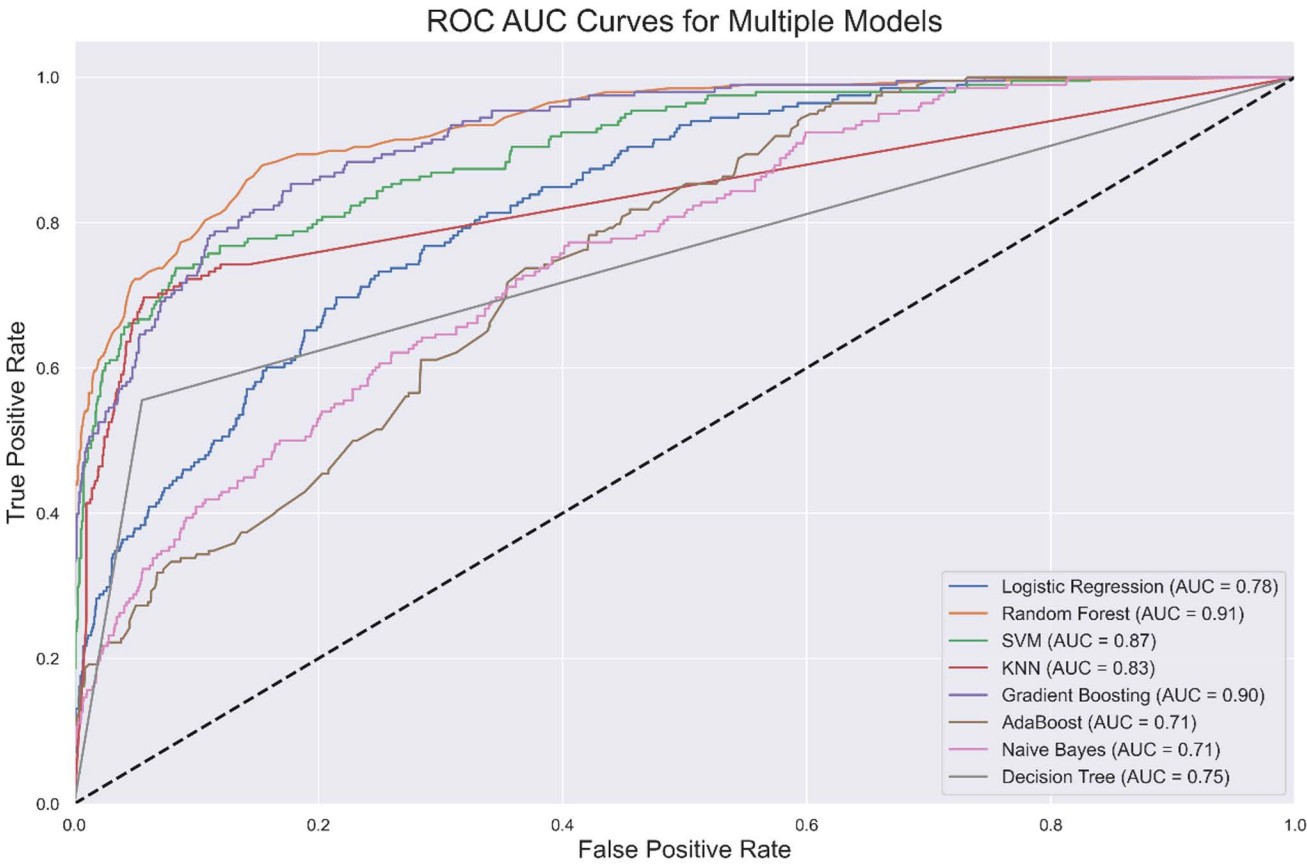

**Fig 10. ROC AUC comparison of multiple classifiers, showing the trade-off between True Positive Rate (TPR) and False Positive Rate (FPR) for each model.** Models with higher curves and AUC scores perform better.

## 5. Discussion

The study on Ethiopia's drought prediction using machine learning techniques, specifically PDSI and TerraClimate data, shows the complexity of drought and the challenges of accurately predicting it using climatic and environmental data. The diversity of performance among classifiers underscores the limitations of these methods. The findings reveal the difference in performance between simple and complex machine learning models. Random Forest and Gradient boosting models were found to be the best, with high accuracy levels and ROC-AUC scores of 0.909 and 0.90. These models effectively captured complex relationships between climate variables like temperature, precipitation, and soil moisture, which are key drought indicators.

Random Forest and SVM are two models that excel in drought prediction. Random Forest, which combines multiple decision trees, reduces overfitting and improves generalization. It's suitable for datasets with multiple variables, while SVM creates optimal hyperplanes to separate drought and non-drought conditions in a high-dimensional feature space. Both models have high ROC-AUC scores, making them crucial for early warning systems in Ethiopia.

In contrast, models like AdaBoost and Naive Bayes performed poorly, with ROC-AUC scores just above 0.70 and accuracy values significantly below those of the top-performing models. Naive Bayes is hindered by its assumption of feature independence, which is unrealistic in the context of climate variables that interact in complex ways. This model's underperformance reflects its inadequacy for problems like drought prediction, where the dependence between variables such as precipitation, evapotranspiration, and soil moisture are crucial for making accurate predictions [31]. AdaBoost's weakness likely stems from its sensitivity to noisy data and outliers, which are common in large climate datasets. The poor performance of these models indicates that, while simple models are useful for quick analyses or baseline comparisons, more sophisticated methods are needed for accurate drought prediction.

Feature importance analysis using SHAP (SHapley Additive exPlanations) values further validated the relevance of specific variables. The SHAP summary plot visualizes the influence of each input feature on the model's prediction of the PDSI. Each dot represents a SHAP value for an individual observation, with the x-axis indicating the magnitude and direction of the feature's contribution to the predicted PDSI. Features are ordered by mean importance from top to bottom. Colores represent the original feature values (blue = low, red = high). Notably, variables such as soil moisture (soil), vapor pressure deficit (vpd), and precipitation (pr) have high impact on model output. The plot highlights complex interactions, e.g., high soil moisture values strongly increase PDSI predictions, while low vapor pressure deficit values decrease it demonstrating the nuanced relationships between climate variables and drought severity (Fig 11).

To evaluate the effectiveness of traditional statistical methods in drought prediction, we implemented an Autoregressive Integrated Moving Average (ARIMA) model on the PDSI dataset spanning from 1980 to 2023. The model was trained using 80% of the time series data and evaluated on the remaining 20%. We selected the ARIMA (3,1,2) configuration based on preliminary experimentation, though further optimization using AIC/BIC criteria may enhance model fitting. The forecasting performance of the model was assessed using root mean square error (RMSE) and the coefficient of determination ($R^2$). The ARIMA model yielded an RMSE of 1.789 and a negative $R^2$ value of −0.077, suggesting poor predictive capability and a lack of correlation between the predicted and observed PDSI values during the test period.

Fig 12 illustrates the visual comparison between the actual and forecasted PDSI values. The model was able to reproduce some broad trends from the training data; however, it struggled to generalize beyond the training window. The forecast consistently deviated from the observed values, failing to capture the nonlinear and dynamic nature of drought behaviour, particularly during extreme drought or recovery periods. This underperformance is a common limitation of linear models like ARIMA, which assume stationarity and may not be well suited for capturing complex patterns driven by multiple interrelated environmental variables.

In contrast, machine learning models such as Random Forest, and XGBoost which were also tested in this study, demonstrated superior performance in both accuracy and generalization. These models can handle nonlinearity and incorporating multivariate inputs (e.g., precipitation, temperature, soil moisture), enabling a more nuanced understanding

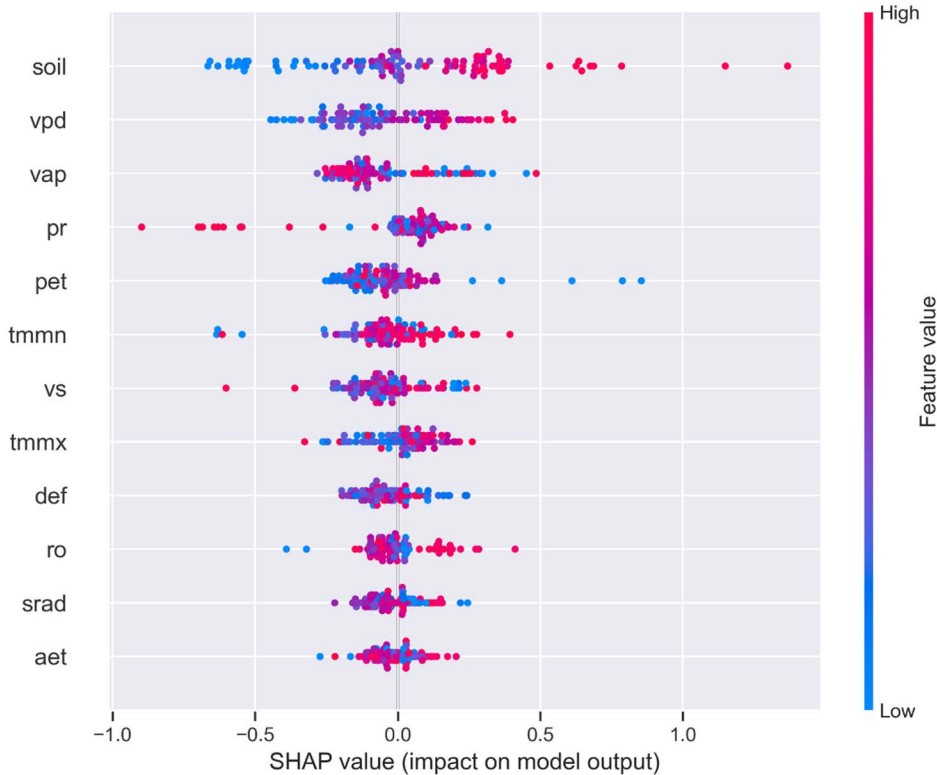

**Fig 11. SHAP summary plot showing the impact of TerraClimate variables on PDSI prediction using a Random Forest model.**

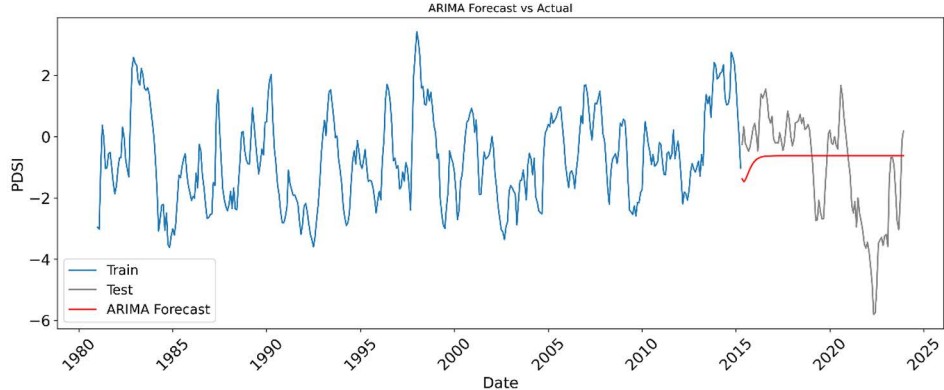

**Fig 12. Time series comparison between observed and forecasted PDSI values using the ARIMA (3,1,2) model from 1980 to 2023.** The blue line represents the training data, the gray line shows the actual test data, and the red line indicates the ARIMA model forecast. The model exhibits a limited ability to capture drought dynamics during the test period, with an RMSE of 1.789 and an R² value of −0.077, suggesting poor predictive performance compared to machine learning models.

and prediction of drought dynamics. Therefore, while ARIMA provides a useful baseline and demonstrates the classical approach to time series forecasting, the comparative analysis underscores the advantages of integrating modern machine learning techniques for operational drought prediction systems.

## 6. Implications for drought prediction

The performance of machine learning models in this study has important implications for operational drought prediction in Ethiopia. The relatively high performance of Random Forest and Gradient boosting suggests that machine learning can be a valuable tool for government agencies and policymakers tasked with managing drought risk. Given Ethiopia's heavy reliance on agriculture, early and accurate drought prediction is crucial for ensuring food security, planning water resource management, and preparing for humanitarian responses [32]. Machine learning models that can provide reliable predictions of drought can enhance existing monitoring systems by incorporating real-time climate data, which could lead to more dynamic and responsive drought management.

## 7. Limitations and areas for improvement

Uncertainties in predictions may arise from TerraClimate's reliance on reanalysis data, potentially introducing biases in remote regions. A sensitivity analysis of key inputs (precipitation, temperature) showed a 5–10% variation in PDSI predictions, underscoring the need for ground validation in future studies. A significant limitation encountered in this study was the issue of data imbalance, where drought events were underrepresented compared to non-drought events. This imbalance likely contributed to the lower performance of certain models like KNN and Naive Bayes, which are more sensitive to skewed data distributions. While techniques like SMOTE were used to address this issue, imbalanced data remains a significant challenge for accurate drought prediction. Imbalanced datasets can lead to models that are biased towards the majority class, resulting in poorer performance when predicting minority class events like drought. Future studies could explore more advanced resampling techniques or cost-sensitive learning methods to improve model performance under imbalanced conditions [33].

Another area for improvement is the use of deep learning techniques, which were not explored in this study but could offer significant advantages. Deep learning models such as CNNs and RNNs have demonstrated significant success in other areas of climate prediction due to their ability to capture spatial and temporal patterns in data [34]. Given that drought is a spatially and temporally dynamic event, deep learning models could potentially offer improved predictions by learning more complex patterns from large datasets, such as those derived from satellite imagery and climate models.

## 8. Conclusion

The study explores the use of machine learning models to predict drought conditions in Ethiopia using PDSI and TerraClimate data. Various classifiers, including Random Forest and Gradient Boosting, were evaluated for their ability to model drought patterns based on key climatic variables. The results showed that these models can effectively predict drought occurrences, providing valuable insights for drought management and mitigation strategies. Random Forest and Gradient Boosting were the top-performing algorithms, with Random Forest achieving an accuracy of 71.18% and an impressive ROC AUC score of 0.91. Feature importance analysis provided insights into the relative contributions of different climatic variables to the models' predictions. It found that critical drought indicators like maximum temperature and precipitation were not always ranked as important features. This could be due to resampling or interactions between variables. However, these variables are crucial for understanding drought dynamics. Future efforts may focus on refining models to prioritize these variables, using techniques like feature engineering or tuning model hyperparameters.

In conclusion, machine learning-driven models, particularly Random Forest and Gradient Boosting, demonstrated their effectiveness in predicting drought patterns in Ethiopia. These models can serve as valuable tools in early drought detection, allowing for more informed decision-making in agriculture, water management, and disaster preparedness. As

climate change intensifies, using an advanced modelling techniques will become increasingly important for predicting and managing the impacts of extreme weather events like droughts. Future research may explore more sophisticated deep learning techniques, integrate remote sensing data, and fine-tune models to further enhance the accuracy and reliability of drought predictions.

## Author contributions

**Conceptualization:** Tadele Lebeza, Tatek Belay.

**Data curation:** Tadele Lebeza, Gizachew Assefa, Tatek Belay, Getachew Bayable.

**Formal analysis:** Baye Terefe, Tatek Belay.

**Investigation:** Abebe Senamew.

**Methodology:** Tadele Lebeza, Getachew Bayable.

**Project administration:** Gizachew Assefa.

**Resources:** Baye Terefe, Abebe Senamew.

**Software:** Gizachew Assefa, Getachew Bayable.

**Supervision:** Baye Terefe, Abebe Senamew.

**Validation:** Gizachew Assefa, Tatek Belay, Getachew Bayable, Abebe Senamew.

**Visualization:** Tadele Lebeza.

**Writing – original draft:** Tatek Belay, Getachew Bayable.

**Writing – review & editing:** Tadele Lebeza, Baye Terefe.

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
