## [Decision Letter · Decision Letter 0]

PONE-D-24-44104Machine Learning-Driven Drought Prediction Using Palmer Drought Severity Index and TerraClimate Data over EthiopiaPLOS ONE

Dear Dr. Lebeza,

Thank you for submitting your manuscript to PLOS ONE. After careful consideration, we feel that it has merit but does not fully meet PLOS ONE’s publication criteria as it currently stands. Therefore, we invite you to submit a revised version of the manuscript that addresses the points raised during the review process.

**ACADEMIC EDITOR: ** We appreciate your contribution to the field. Based on the reviewers' feedback, we have identified key areas that require substantial revision before the manuscript can be reconsidered for publication.

We request that you submit a **detailed response letter** alongside your revised manuscript, addressing each reviewer comment point by point. Please clearly indicate the changes made and provide justifications where necessary.

We look forward to receiving your revised manuscript.

Kind regards,

Suresh Devaraj

Academic Editor

PLOS ONE

2. For studies involving third-party data, we encourage authors to share any data specific to their analyses that they can legally distribute. PLOS recognizes, however, that authors may be using third-party data they do not have the rights to share. When third-party data cannot be publicly shared, authors must provide all information necessary for interested researchers to apply to gain access to the data. (https://journals.plos.org/plosone/s/data-availability#loc-acceptable-data-access-restrictions)

4. We note you have included a table to which you do not refer in the text of your manuscript. Please ensure that you refer to Table 1 in your text; if accepted, production will need this reference to link the reader to the Table.

5. Please ensure that you refer to Figures 1, 6, 9, 10 in your text as, if accepted, production will need this reference to link the reader to the figure.

Additional Editor Comments (if provided):

Reviewers' comments:

Reviewer's Responses to Questions

**Comments to the Author**

1. Is the manuscript technically sound, and do the data support the conclusions?

Reviewer #1: Yes

Reviewer #2: Yes

2. Has the statistical analysis been performed appropriately and rigorously? 

Reviewer #1: Yes

Reviewer #2: Yes

3. Have the authors made all data underlying the findings in their manuscript fully available?

Reviewer #1: Yes

Reviewer #2: Yes

4. Is the manuscript presented in an intelligible fashion and written in standard English?

Reviewer #1: Yes

Reviewer #2: Yes

5. Review Comments to the Author

Reviewer #1: MANUSCRIPT ID: PONE-D-24-44104

MANUSCRIPT TITLE: Machine Learning-Driven Drought Prediction Using Palmer Drought Severity Index and Terra Climate Data over Ethiopia

REVIEWER COMMENT

The manuscript entitled “Machine Learning-Driven Drought Prediction Using Palmer Drought Severity Index and Terra Climate Data over Ethiopia”. The information provided in this manuscript is beneficial for researchers and academia, but this study has some limitation as the provided information is only for a specific area. There is always room for improvement, and I have suggested some minor revisions.

Title- Please review as per your aim and objectives as not reflecting the true sense of article

Abstract: results- Mention numerical values for better understanding- The abstract says the study uses advanced machine learning techniques for example logistic regression (abstract line 23), but this is statistical analysis technique as well so make it clear how these techniques serve in this study.

TerraClimate data - The author mentioned in introduction and methodology several times that they use high resolutions climate data please mentioned resolution range for clarification.

Data source: Could the author explain how they extract climate data through google earth engine the study here uses precipitation, temperature and soil moisture data.

Figures 4,5 and 6 need more clarification regarding value, time series and variable explanation.

• Please avoid repetition-

• Please check reference style throughout MS

• Recheck Legends description is as per figure number and discussion-

Reviewer #2: Review Comments

Here is a detailed review of research article, "Machine Learning-Driven Drought Prediction Using Palmer Drought Severity Index and Terra Climate Data over Ethiopia."

Overall Assessment

The study is well-structured and presents a significant contribution to drought prediction using machine learning. The use of multiple classifiers and evaluation metrics provides a comprehensive analysis. However, there are a few areas where improvements could enhance the clarity, robustness, and impact of the study.

Major Comments

Justification for Model Selection

• The study uses multiple machine learning models, but a stronger justification for selecting these specific models (e.g., Logistic Regression, Random Forest, SVM, KNN, etc.) should be included.

• Consider briefly discussing why deep learning models (CNN, LSTM) were not included, given their growing use in environmental modeling.

Handling of Imbalanced Data

• The manuscript mentions using SMOTE and manual duplication to address data imbalance, but this method might lead to artificial patterns in the dataset.

• A comparison with other balancing techniques (such as class-weighting or cost-sensitive learning) would be beneficial.

Feature Importance Analysis

• The study mentions that key drought predictors such as temperature and precipitation were not always ranked highly. This contradicts conventional knowledge and should be explained more clearly.

• Conducting SHAP (SHapley Additive Explanations) analysis could offer deeper insights into feature contributions.

Uncertainty and Limitations

• There is little discussion on the uncertainties associated with machine learning predictions, including possible biases in the TerraClimate dataset.

• A sensitivity analysis of input variables could improve confidence in the model outputs.

Comparison with Traditional Models

• Since traditional statistical models (e.g., autoregressive models, time-series forecasting) are commonly used in drought prediction, it would strengthen the paper to compare at least one traditional model with the machine learning models.

Minor Comments

1. Clarity in Model Performance Discussion

• The discussion on Random Forest and Gradient Boosting being the best models should explicitly state their precision, recall, and F1-score rather than focusing mainly on accuracy and ROC-AUC.

2. Data Preprocessing Details

• More details on how missing data was handled (linear interpolation was mentioned, but were other methods like KNN imputation tested?) would strengthen the methodology.

3. Figures & Tables

• Ensure that all figures are high resolution and clearly labeled (some descriptions refer to figures without clear reference in the text).

• Consider adding a table comparing model performance metrics for quick readability.

4. Language & Grammar

• Minor grammatical errors and awkward phrasing exist (e.g., “The dataset shows varying degrees of drought and wet conditions, with Near Normal being the most common category” → consider rewording for clarity).

• Proofreading for conciseness and academic tone is recommended.

6. PLOS authors have the option to publish the peer review history of their article (what does this mean? ). If published, this will include your full peer review and any attached files.

**Do you want your identity to be public for this peer review?** For information about this choice, including consent withdrawal, please see our Privacy Policy .

Reviewer #1: **Yes: ** Md Abdullah Al Mamun

Department of Geography

Hong Kong Baptist University

Institutional email -21482764@life.hkbu.edu.hk

Reviewer #2: **Yes: ** Dr Mahesh Palakuru

---

## [Author Response · Author response to Decision Letter 1]

26 Apr 2025

Response to Editor and Reviewers

Manuscript ID: PONE-D-24-44104

Title: Machine Learning-Driven Drought Prediction Using Palmer Drought Severity Index and TerraClimate Data over Ethiopia

Dear Editor and Reviewers,

Thank you for the opportunity to revise and resubmit our manuscript to PLOS ONE. We would like to thank for the constructive feedback provided by the reviewers and the editorial team, which has helped us improve the clarity, rigor, and impact of our study. In this revised submission, we have addressed all editorial requirements and responded to each reviewer’s comments in detail. The sections highlighted in red represent our responses to both the editor and reviewers' feedback and indicate the corresponding revisions made in the main manuscript. All suggested changes have been incorporated into the revised manuscript, and we have uploaded all additional files as requested.

Response to Editorial Requirements

Response: We have carefully revised our manuscript to ensure full compliance with PLOS ONE’s formatting and style guidelines as requested. All files have been renamed according to PLOS ONE’s file naming conventions. We have reformatted the manuscript using the PLOS ONE style templates provided. The main text of the manuscript follows the structure and formatting style outlined in the Main Body Sample Template. The title page, authorship, and affiliations have been revised to match the formatting in the Title, Authors, and Affiliations Template.

2. Third-Party Data Availability

Response: We acknowledge PLOS ONE’s policy on data availability and the encouragement to share third-party data where legally permissible. The data underlying the results presented in this study are obtained from TerraClimate via Google Earth Engine. TerraClimate provides high-resolution (1/24°, ~4-km) monthly climate and hydrological data from 1958 to the present. These datasets are publicly accessible on Google Earth Engine (https://earthengine.google.com) by searching for the TerraClimate dataset. Verification of permission is not applicable as TerraClimate is an open-access resource. The authors did not receive any special privileges in accessing the data that other researchers would not have; access requires only a free Google Earth Engine account. For further assistance, researchers may contact Google Earth Engine support via their official website (https://earthengine.google.com/support).

3. Copyright Concerns for Figure 1

We note that Figure 1 in your submission contain [map/satellite] images which may be copyrighted. All PLOS content is published under the Creative Commons Attribution License (CC BY 4.0), which means that the manuscript, images, and Supporting Information files will be freely available online, and any third party is permitted to access, download, copy, distribute, and use these materials in any way, even commercially, with proper attribution. For these reasons, we cannot publish previously copyrighted maps or satellite images created using proprietary data, such as Google software (Google Maps, Street View, and Earth). For more information, see our copyright guidelines: http://journals.plos.org/plosone/s/licenses-and-copyright.

In the figure caption of the copyrighted figure, please include the following text: “Reprinted from [ref] under a CC BY license, with permission from [name of publisher], original copyright [original copyright year].

2. If you are unable to obtain permission from the original copyright holder to publish these figures under the CC BY 4.0 license or if the copyright holder’s requirements are incompatible with the CC BY 4.0 license, please either i) remove the figure or ii) supply a replacement figure that complies with the CC BY 4.0 license. Please check copyright information on all replacement figures and updatea the figure caption with source information. If applicable, please specify in the figure caption text when a figure is similar but not identical to the original image and is therefore for illustrative purposes only.

Response: Thank you for the feedback. We have removed the base map from Figure 1 as requested. The revised figure now only incorporates freely available precipitation data from the TerraClimate dataset, which we accessed using Google Earth Engine, along with other publicly available datasets. No copyrighted or proprietary base maps or satellite imagery were used. We have uploaded the revised figure with the resubmission and ensured it meets PLOS ONE’s copyright standards.

4. We note you have included a table to which you do not refer in the text of your manuscript. Please ensure that you refer to Table 1 in your text; if accepted, production will need this reference to link the reader to the Table.

Response: Thank you for noting the omission of a reference to Table 1 in the text. We have added a reference in the "Materials and Methods" section under 3.1. Data Collection (TerraClimate Data subsection) as follows.

The TerraClimate dataset includes a range of climate and water balance variables essential for analysing drought dynamics (see Table 1 for variable descriptions, units, and scales).

5. Please ensure that you refer to Figures 1, 6, 9, 10 in your text as, if accepted, production will need this reference to link the reader to the figure.

Response: We apologize for the oversight in not referencing Figures 1, 6, 9, and 10 explicitly in the text. Figure 1 is added in Study Area Description section. The study area encompasses Ethiopia, located in the Horn of Africa (Figure 1). Figure 6 is added in Data Preprocessing (EDA subsection). Boxplots were used to identify and manage outliers, with comparisons before and after outlier removal shown in Figure 6. Figure 9 is added in Results and Discussion (Hyperparameter Tuning subsection). ROC curves for multiple classifiers across PDSI classes are presented in Figure 9, illustrating their discriminative abilities. Figure 10 Added in Results and Discussion (Discussion subsection). A comparison of ROC AUC scores across all models is shown in Figure 10, highlighting the superior performance of Random Forest and Gradient Boosting.

Response to Reviewer Comments

Reviewer #1

General Comment:

We thank Reviewer #1 for the positive assessment and valuable suggestions. We agree that while our study focuses on Ethiopia, its methodology could inspire broader applications. Below, we address the specific points raised.

1. Title- Please review as per your aim and objectives as not reflecting the true sense of article

Response: We have reconsidered the title to better reflect the study’s aim and objectives. The original title emphasized machine learning and data sources but could be refined for clarity. The revised title is:

“Machine Learning-Based Drought Prediction Using Palmer Drought Severity Index and TerraClimate Data in Ethiopia.”

This maintains focus on the methodology, data, and geographic scope while aligning with the study’s objectives.

2. Abstract: results- Mention numerical values for better understanding- The abstract says the study uses advanced machine learning techniques for example logistic regression (abstract line 23), but this is statistical analysis technique as well so make it clear how these techniques serve in this study.

Response: We have added key numerical results to the abstract for better understanding and clarified the role of logistic regression as a machine learning technique. This distinguishes Logistic Regression’s role while emphasizing its machine learning application.

The revised abstract now includes:

“We employed several classifiers Logistic Regression, Support Vector Machine (SVM), k-Nearest Neighbors (KNN), Decision Tree, Random Forest, Gradient Boosting, Naive Bayes, AdaBoost, and XGBoost with Logistic Regression serving as a baseline statistical approach for comparison. To address data imbalance across drought classes, we applied a hybrid resampling method combining manual upsampling and SMOTE. Hyperparameter tuning was conducted using grid search and cross-validation. Random Forest outperformed all models, achieving an accuracy of 71.18%, F1-score of 0.71, and ROC AUC of 0.9000. Gradient Boosting and SVM also performed well with ROC AUC values of 0.8982 and 0.8681, respectively..”

3. TerraClimate data - The author mentioned in introduction and methodology several times that they use high resolutions climate data please mentioned resolution range for clarification.

Response: We appreciate the request for clarification on resolution. TerraClimate data has a spatial resolution of 1/24° (~4-km), which we have now specified in the 3.1. Data Collection (TerraClimate Data subsection):

"The TerraClimate dataset offers global, monthly gridded data at a high spatial resolution of 1/24° (~4-km), covering climate and water balance variables from 1958 to the present…"

4. Data source: Could the author explain how they extract climate data through google earth engine the study here uses precipitation, temperature and soil moisture data.

Response: Thank you for your insightful comment. We have elaborated on the data extraction process in "3.1. Data Collection" (TerraClimate Data subsection):

In this study, we extracted the climate data using the TerraClimate dataset available on Google Earth Engine (GEE) (ID: IDAHO_EPSCOR/TERRACLIMATE) in https://earthengine.google.com/. This dataset provides high-resolution (approximately 4-km) monthly climate and climatic water balance data. We followed these key steps for data extraction through GEE. We used the IDAHO_EPSCOR/TERRACLIMATE dataset, which includes monthly precipitation (pr), maximum temperature (tmmx), minimum temperature (tmmn), and soil moisture (soil). We defined the geographic extent of our study area using a shapefile (uploaded to GEE Assets) and clipped the data accordingly. We filtered the TerraClimate image collection by date to match the study period. For each selected variable, we extracted relevant bands and computed spatial means over the study area to generate time series data. The processed data were exported as CSV files from GEE to Google Drive and subsequently used for model training and evaluation.

5. Figures 4,5 and 6 need more clarification regarding value, time series and variable explanation.

Response: We have revised the captions for Figures 4, 5, and 6 to provide more detail:

• Figure 4: Time series analysis of TerraClimate variables (e.g., precipitation, maximum temperature, soil moisture) from 1981 to 2023, illustrating seasonal trends and patterns over time.

• Figure 5: Distribution analysis of PDSI from 1981 to 2023, featuring the Probability Density Function (PDF, left) and Cumulative Distribution Function (CDF, right). The red dashed line indicates the mean PDSI (-0.12), with blue dashed lines showing one standard deviation (±1.85).

• Figure 6: Comparison of boxplots for TerraClimate variables (e.g., PET, SRAD) before (left) and after (right) outlier removal, showing reduced variability post-capping of extreme values.

6. Additional Suggestions

• Please avoid Repetition:

Response: We have reviewed the manuscript to eliminate redundancy, e.g., consolidating repeated mentions of TerraClimate’s utility in the Introduction and Methods.

• Please check reference style throughout MS

Response: We have ensured consistency with PLOS ONE’s style, correcting in-text citations and reference formatting.

• Recheck Legends description is as per figure number and discussion

Response: All figure captions have been cross-checked to match their descriptions and discussion references.

Reviewer #2

General Comment:

We thank Reviewer #2 for the thorough review and insightful suggestions, which have significantly strengthened our manuscript. Below, we address each point.

Overall Assessment

The study is well-structured and presents a significant contribution to drought prediction using machine learning. The use of multiple classifiers and evaluation metrics provides a comprehensive analysis. However, there are a few areas where improvements could enhance the clarity, robustness, and impact of the study.

Major Comments:

1. Justification for Model Selection:

• The study uses multiple machine learning models, but a stronger justification for selecting these specific models (e.g., Logistic Regression, Random Forest, SVM, KNN, etc.) should be included.

• Consider briefly discussing why deep learning models (CNN, LSTM) were not included, given their growing use in environmental modelling.

Response: We have added a paragraph in 3.3. Machine Learning Models for Classification (Model Selection subsection).

“Model selection was guided by the diversity of algorithmic approaches and their proven efficacy in environmental modelling. The chosen models included Logistic Regression, Random Forest, Support Vector Machine (SVM) (Gandhi, 2018), K-Nearest Neighbors (KNN), Gradient Boosting, , Naive Bayes, and Decision Tree (Fratello & Tagliaferri, 2018). Logistic Regression was chosen for its simplicity and interpretable base line, providing a probabilistic approach to class membership. Random Forest was selected for its ensemble method that combines multiple decision trees to enhance classification accuracy and robustness (Belgiu & Drăgu, 2016). SVM was included due to its effectiveness in handling complex decision boundaries using various kernel functions. KNN was used for its non-parametric approach, which classifies data based on the majority vote of nearest neighbors, capturing local patterns. Gradient Boosting and AdaBoost were selected for their ensemble techniques that improve classification performance by aggregating multiple weak classifiers AdaBoost (Callens et al., 2020; Diantika et al., 2021). Naive Bayes was chosen for its efficiency and simplicity, based on Bayes’ theorem with strong independence assumptions. Finally, Decision Tree was included for its ability to provide clear, interpretable decision rules by splitting data based on feature values. Deep learning mod

---

## [Decision Letter · Decision Letter 1]

Machine Learning-Based Drought Prediction Using Palmer Drought Severity Index and TerraClimate Data in Ethiopia

PONE-D-24-44104R1

Dear Dr. Lebeza,

We’re pleased to inform you that your manuscript has been judged scientifically suitable for publication and will be formally accepted for publication once it meets all outstanding technical requirements.

Kind regards,

Suresh Devaraj

Academic Editor

PLOS ONE

Additional Editor Comments (optional):

Reviewers' comments:

Reviewer's Responses to Questions

**Comments to the Author**

1. If the authors have adequately addressed your comments raised in a previous round of review and you feel that this manuscript is now acceptable for publication, you may indicate that here to bypass the “Comments to the Author” section, enter your conflict of interest statement in the “Confidential to Editor” section, and submit your "Accept" recommendation.

Reviewer #1: All comments have been addressed

2. Is the manuscript technically sound, and do the data support the conclusions?

Reviewer #1: Yes

3. Has the statistical analysis been performed appropriately and rigorously? 

Reviewer #1: Yes

4. Have the authors made all data underlying the findings in their manuscript fully available?

Reviewer #1: No

5. Is the manuscript presented in an intelligible fashion and written in standard English?

Reviewer #1: Yes

6. Review Comments to the Author

Reviewer #1: The author has thoroughly addressed all the comments and suggestions provided during the review process. As a reviewer, I acknowledge the author's efforts in rectifying the issues identified in the manuscript. For final suggestions, I recommend ensuring that all technical abbreviations, such as ROC and AUC, are fully elaborated and accompanied by brief explanations to enhance the manuscript's accessibility for a broader audience. I extend my best wishes to the author for the success of their work.

7. PLOS authors have the option to publish the peer review history of their article (what does this mean? ). If published, this will include your full peer review and any attached files.

**Do you want your identity to be public for this peer review?** For information about this choice, including consent withdrawal, please see our Privacy Policy .

Reviewer #1: **Yes: ** Md Abdullah Al Mamun

---

## [Editor Report · Acceptance letter]

PONE-D-24-44104R1

PLOS ONE

Dear Dr. Lebeza,

I'm pleased to inform you that your manuscript has been deemed suitable for publication in PLOS ONE. Congratulations! Your manuscript is now being handed over to our production team.

Kind regards,

on behalf of

Dr. Suresh Devaraj

Academic Editor

PLOS ONE